# A Greedy Approach for
# Budgeted Maximum Inner Product Search

**Hsiang-Fu Yu**[*]
Amazon Inc.
rofuyu@cs.utexas.edu

**Cho-Jui Hsieh**
University of California, Davis
chohsieh@ucdavis.edu

**Qi Lei**
The University of Texas at Austin
leiqi@ices.utexas.edu

**Inderjit S. Dhillon**
The University of Texas at Austin
inderjit@cs.utexas.edu

## Abstract

Maximum Inner Product Search (MIPS) is an important task in many machine learning applications such as the prediction phase of low-rank matrix factorization models and deep learning models. Recently, there has been substantial research on how to perform MIPS in sub-linear time, but most of the existing work does not have the flexibility to control the trade-off between search efficiency and search quality. In this paper, we study the important problem of MIPS with a computational budget. By carefully studying the problem structure of MIPS, we develop a novel Greedy-MIPS algorithm, which can handle budgeted MIPS by design. While simple and intuitive, Greedy-MIPS yields surprisingly superior performance compared to state-of-the-art approaches. As a specific example, on a candidate set containing half a million vectors of dimension 200, Greedy-MIPS runs 200x faster than the naive approach while yielding search results with the top-5 precision greater than 75%.

## 1 Introduction

In this paper, we study the computational issue in the prediction phase for many embedding based models such as matrix factorization and deep learning models in recommender systems, which can be mathematically formulated as a Maximum Inner Product Search (MIPS) problem. Specifically, given a large collection of $n$ candidate vectors: $\mathcal{H} = \left\{ \boldsymbol{h}_j \in \mathbb{R}^k : 1, \dots, n \right\}$ and a query vector $\boldsymbol{w} \in \mathbb{R}^k$, MIPS aims to identify a subset of candidates that have top largest inner product values with $\boldsymbol{w}$. We also denote $H = [\boldsymbol{h}_1, \dots, \boldsymbol{h}_j, \dots, \boldsymbol{h}_n]^\top$ as the candidate matrix. A naive linear search procedure to solve MIPS for a given query $\boldsymbol{w}$ requires $O(nk)$ operations to compute $n$ inner products and $O(n \log n)$ operations to obtain the sorted ordering of the $n$ candidates.

Recently, MIPS has drawn a lot of attention in the machine learning community due to its wide applicability, such as the prediction phase of embedding based recommender systems [6, 7, 10]. In such an embedding based recommender system, each user $i$ is associated with a vector $\boldsymbol{w}_i$ of dimension $k$, while each item $j$ is associated with a vector $\boldsymbol{h}_j$ of dimension $k$. The interaction (such as preference) between a user and an item is modeled by $\boldsymbol{w}_i^T \boldsymbol{h}_j$. It is clear that identifying top-ranked items in such a system for a user is exactly a MIPS problem. Because both the number of users (the number of queries) and the number of items (size of vector pool in MIPS) can easily grow to millions, a naive linear search is extremely expensive; for example, to compute the preference for all $m$ users over $n$ items with latent embeddings of dimension $k$ in a recommender system requires at least $O(mnk)$ operations. When both $m$ and $n$ are large, the prediction procedure is extremely time consuming; it is even *slower* than the training procedure used to obtain the $m + n$ embeddings, which

---

[*]Work done while at the University of Texas at Austin.

costs only $O(|\Omega|k)$ operations per iteration, where $|\Omega|$ is number of observations and is much smaller than $mn$. Taking the yahoo-music dataset as an example, $m = 1M$, $n = 0.6M$, $|\Omega| = 250M$, and $mn = 600B \gg 250M = |\Omega|$. As a result, the development of efficient algorithms for MIPS is needed in large-scale recommender systems. In addition, MIPS can be found in many other machine learning applications, such as the prediction for a multi-class or multi-label classifier [16, 17], an object detector, a structure SVM predicator, or as a black-box routine to improve the efficiency of learning and inference algorithm [11]. Also, the prediction phase of neural network could also benefit from a faster MIPS algorithm: the last layer of NN is often a dense fully-connected layer, so finding the label with maximum score becomes a MIPS problem with *dense vectors* [6].

There is a recent line of research on accelerating MIPS for large $n$, such as [2, 3, 9, 12–14]. However, most of them do not have the flexibility to control the trade-off between search efficiency and search quality in the prediction phase. In this paper, we consider the budgeted MIPS problem, which is a generalized version of the standard MIPS with a computation budget: *how to generate a set of top-ranked candidates under a given* **budget** *on the number of inner products one can perform.* By carefully studying the problem structure of MIPS, we develop a novel Greedy-MIPS algorithm, which handles budgeted MIPS by design. While simple and intuitive, Greedy-MIPS yields surprisingly superior performance compared to existing approaches.

**Our Contributions:**
- We develop Greedy-MIPS, which is a novel algorithm without any nearest neighbor search reduction that is essential in many state-of-the-art approaches [2, 12, 14].
- We establish a sublinear time theoretical guarantee for Greedy-MIPS under certain assumptions.
- Greedy-MIPS is orders of magnitudes faster than many state-of-the-art MIPS approaches to obtain a desired search performance. As a specific example, on the yahoo-music data sets with $n = 624,961$ and $k = 200$, Greedy-MIPS runs 200x faster than the naive approach and yields search results with the top-5 precision more than $75\%$, while the search performance of other state-of-the-art approaches under the similar speedup drops to less than $3\%$ precision.
- Greedy-MIPS supports MIPS with a budget, which brings the ability to control of the trade-off between computation efficiency and search quality in the prediction phase.

## 2   Existing Approaches for Fast MIPS

Because of its wide applicability, several algorithms have been proposed for efficient MIPS. Most of existing approaches consider to reduce the MIPS problem to the nearest neighbor search problem (NNS), where the goal is to identify the nearest candidates of the given query, and apply an existing efficient NNS algorithm to solve the reduced problem. [2] is the first MIPS work which adopts such a MIPS-to-NNS reduction. Variants MIPS-to-NNS reduction are also proposed in [14, 15]. Experimental results in [2] show the superiority of the NNS reduction over the traditional branch-and-bound search approaches for MIPS [9, 13]. After the reduction, there are many choices to solve the transformed NNS problem, such as locality sensitive hashing scheme (LSH-MIPS) considered in [12, 14, 15], PCA-tree based approaches (PCA-MIPS) in [2], or K-Means approaches in [1].

Fast MIPS approaches with sampling schemes have become popular recently. Various sampling schemes have been proposed to handle MIPS problem with different *constraints*. The idea of the sampling-based MIPS approach is first proposed in [5] as an approach to perform approximate matrix-matrix multiplications. Its applicability on MIPS problems is studied very recently [3]. The idea behind a sampling-based approach called Sample-MIPS, is about to design an efficient sampling procedure such that the $j$-th candidate is selected with probability $p(j)$: $p(j) \sim \boldsymbol{h}_j^\top \boldsymbol{w}$. In particular, Sample-MIPS is an efficient scheme to sample $(j, t) \in [n] \times [k]$ with the probability $p(j, t)$: $p(j, t) \sim h_{jt} w_t$. Each time a pair $(j, t)$ is sampled, we increase the count for the $j$-th item by one. By the end of the sampling process, the spectrum of the counts forms an estimation of $n$ inner product values. Due to the nature of the sampling approach, it can only handle the situation where all the candidate vectors and query vectors are *nonnegative*.

Diamond-MSIPS, a diamond sampling scheme proposed in [3], is an extension of Sample-MIPS to handle the maximum **squared** inner product search problem (MSIPS) where the goal is to identify candidate vectors with largest values of $(\boldsymbol{h}_j^\top \boldsymbol{w})^2$. However, the solutions to MSIPS can be very different from the solutions to MIPS in general. For example, if all the inner product values are negative, the ordering for MSIPS is the exactly reverse ordering induced by MIPS. Here we can see that the applicability of both Sample-MIPS and Diamond-MSIPS to MIPS is very limited.

# 3 Budgeted MIPS

The core idea behind the fast approximate MIPS approaches is to trade the search quality for the shorter query latency: the shorter the search latency, the lower the search quality. In most existing fast MIPS approaches, the trade-off depends on the approach-specific parameters such as the depth of the PCA tree in PCA-MIPS or the number of hash functions in LSH-MIPS. Such specific parameters are usually required to construct approach-specific data structures before any query is given, which means that the trade-off is somewhat *fixed* for all the queries. Thus, the computation cost for a given query is fixed. However, in many real-world scenarios, each query might have a different computational budget, which raises the question: *Can we design a MIPS approach supporting the dynamic adjustment of the trade-off in the query phase?*

## 3.1 Essential Components for Fast MIPS

**Before any query request:**
- *Query-Independent Data Structure Construction:* A pre-processing procedure is performed on the entire candidate sets to construct an approach-specific data structure $\mathcal{D}$ to store information about $\mathcal{H}$: the LSH hash tables, space partition trees (e.g., KD-tree or PCA-tree), or cluster centroids.

**For each query request:**
- *Query-dependent **P**re-processing:* In some approaches, a query dependent pre-processing is needed. For example, a vector augmentation is required in all MIPS-to-NNS approaches. In addition, [2] also requires another normalization. $T_P$ is used to denote the time complexity of this stage.
- *Candidate **S**creening:* In this stage, based on the pre-constructed data structure $\mathcal{D}$, an efficient procedure is performed to filter candidates such that only a subset of candidates $\mathcal{C}(\boldsymbol{w}) \subset \mathcal{H}$ is selected. In a naive linear approach, no screening procedure is performed, so $\mathcal{C}(\boldsymbol{w})$ simply contains all the $n$ candidates. For a tree-based structure, $\mathcal{C}(\boldsymbol{w})$ contains all the candidates stored in the leaf node of the query vector. In a sampling-based MIPS approach, an efficient sampling scheme is designed to generate highly possible candidates to form $\mathcal{C}(\boldsymbol{w})$. $T_S$ denotes the computational cost of the screening stage.
- *Candidate **R**anking:* An exact ranking is performed on the selected candidates in $\mathcal{C}(\boldsymbol{w})$ obtained from the screening stage. This involves the computation of $|\mathcal{C}(\boldsymbol{w})|$ inner products and the sorting procedure among these $|\mathcal{C}(\boldsymbol{w})|$ values. The overall time complexity $T_R = O(|\mathcal{C}(\boldsymbol{w})|k + |\mathcal{C}(\boldsymbol{w})| \log|\mathcal{C}(\boldsymbol{w})|)$.

$$\text{The per-query computational cost: } T_Q = T_P + T_S + T_R. \tag{1}$$

It is clear that the candidate screening stage is the *key* component for a fast MIPS approach. In terms of the search quality, the performance highly depends on whether the screening procedure can identify highly possible candidates. Regarding the query latency, the efficiency highly depends on the size of $\mathcal{C}(\boldsymbol{w})$ and how fast to generate $\mathcal{C}(\boldsymbol{w})$. The major difference among various MIPS approaches is the choice of the data structure $\mathcal{D}$ and the screening procedure.

## 3.2 Budgeted MIPS: Problem Definition

Budgeted MIPS is an extension of the standard approximate MIPS problem with a computational budget: how to generate top-ranked candidates under a given **budget** on the number of inner products one can perform. Note that the cost for the candidate ranking ($T_R$) is inevitable in the per-query cost (1). A viable approach for budgeted MIPS must include a screening procedure which satisfies the following requirements:

- the flexibility to control the size of $\mathcal{C}(\boldsymbol{w})$ in the candidate screening stage such that $|\mathcal{C}(\boldsymbol{w})| \leq B$, where $B$ is a given budget, and
- an efficient screening procedure to obtain $\mathcal{C}(\boldsymbol{w})$ in $O(Bk)$ time such that $T_Q = O(Bk + B \log B)$.

As mentioned earlier, most recently proposed MIPS-to-NNS approaches algorithms apply various search space partition data structures or techniques (e.g., LSH, KD-tree, or PCA-tree) designed for NNS to index the candidates $\mathcal{H}$ in the *query-independent pre-processing* stage. As the construction of $\mathcal{D}$ is query independent, both the **search performance** and the **computation cost** are somewhat *fixed* when the construction is done. For example, the performance of a PCA-MIPS depends on the depth of the PCA-tree. Given a query vector $\boldsymbol{w}$, there is no control to the size of $\mathcal{C}(\boldsymbol{w})$ in the candidate generating phase. LSH-based approaches also have the similar issue. There might be some ad-hoc treatments to adjust $\mathcal{C}(\boldsymbol{w})$, it is not clear how to generalize PCA-MIPS and LSH-MIPS in a principled way to handle the situation with a computational budget: how to reduce the size of $\mathcal{C}(\boldsymbol{w})$ under a limited budget and how to improve the performance when a larger budget is given.

Unlike other NNS-based algorithms, the design of Sample-MIPS naturally enables it to support budgeted MIPS for a nonnegative candidate matrix $H$ and a nonnegative query $\boldsymbol{w}$. The more the number of samples, the lower the variance of the estimated frequency spectrum. Clearly, Sample-MIPS has the flexibility to control the size of $\mathcal{C}(\boldsymbol{w})$, and thus is a viable approach for the budgeted MIPS problem. However, Sample-MIPS works only on the situation with non-negative $\mathcal{H}$ and $\boldsymbol{w}$. Diamond-MSIPS has the similar issue.

## 4 Greedy-MIPS

We carefully study the structure of MIPS and develop a simple but novel algorithm called Greedy-MIPS, which handles budgeted MIPS by design. Unlike the recent MIPS-to-NNS approaches, Greedy-MIPS is an approach without any reduction to a NNS problem. Moreover, Greedy-MIPS is a viable approach for the budgeted MIPS problem without the non-negativity limitation inherited in the sampling approaches.

The key component for a fast MIPS approach is the algorithm used in the candidate screening phase. In budgeted MIPS, for any given budget $B$ and query $\boldsymbol{w}$, an *ideal procedure* for the candidate screening phase costs $O(Bk)$ time to generate $\mathcal{C}(\boldsymbol{w})$ which contains the $B$ items with the largest $B$ inner product values over the $n$ candidates in $\mathcal{H}$. The requirement on the time complexity $O(Bk)$ implies that the procedure is independent from $n = |\mathcal{H}|$, the number of candidates in $\mathcal{H}$. One might wonder whether such an ideal procedure exists or not. In fact, designing such an ideal procedure with the requirement to *generate the largest $B$ items* in $O(Bk)$ time is even more challenging than the original budgeted MIPS problem.

**Definition 1.** *The* **rank** *of an item $x$ among a set of items $\mathcal{X} = \{x_1, \ldots, x_{|\mathcal{X}|}\}$ is defined as*

$$\mathbf{rank}(x \mid \mathcal{X}) := \sum\nolimits_{j=1}^{|\mathcal{X}|} \mathbb{I}[x_j \geq x], \tag{2}$$

*where $\mathbb{I}[\cdot]$ is the indicator function. A ranking induced by $\mathcal{X}$ is a function $\pi(\cdot) : \mathcal{X} \to \{1, \ldots, |\mathcal{X}|\}$ such that $\pi(x_j) = \mathbf{rank}(x_j \mid \mathcal{X}) \quad \forall x_j \in \mathcal{X}$.*

One way to store a ranking $\pi(\cdot)$ induced by $\mathcal{X}$ is by a sorted index array $\mathbf{s}[\mathbf{r}]$ of size $|\mathcal{X}|$ such that

$$\pi(x_{\mathbf{s}[1]}) \leq \pi(x_{\mathbf{s}[2]}) \leq \cdots \leq \pi(x_{\mathbf{s}[|\mathcal{X}|]}).$$

We can see that $\mathbf{s}[\mathbf{r}]$ stores the index to the item $x$ with $\pi(x) = r$.

To design an efficient candidate screening procedure, we study the operations required for MIPS: In the simple linear MIPS approach, $nk$ multiplication operations are required to obtain $n$ inner product values $\{\boldsymbol{h}_1^\top \boldsymbol{w}, \ldots, \boldsymbol{h}_n^\top \boldsymbol{w}\}$. We define an *implicit matrix* $Z \in \mathbb{R}^{n \times k}$ as $Z = H \operatorname{diag}(\boldsymbol{w})$, where $\operatorname{diag}(\boldsymbol{w}) \in \mathbb{R}^{k \times k}$ is a matrix with $\boldsymbol{w}$ as it diagonal. The $(j, t)$ entry of $Z$ denotes the multiplication operation $z_{jt} = h_{jt} w_t$ and $\boldsymbol{z}_j = \operatorname{diag}(\boldsymbol{w}) \boldsymbol{h}_j$ denotes the $j$-th row of $Z$. In Figure 1, we use $Z^\top$ to demonstrate the implicit matrix. Note that $Z$ is query dependant, i.e., the values of $Z$ depend on the query vector $\boldsymbol{w}$, and $n$ inner product values can be obtained by taking the column-wise summation of $Z^\top$. In particular, for each $j$ we have $\boldsymbol{h}_j^\top \boldsymbol{w} = \sum_{t=1}^k z_{jt}, \ j = 1, \ldots, n$. Thus, the ranking induced by the $n$ inner product values can be characterized by the *marginal ranking* $\pi(j|\boldsymbol{w})$ defined on the implicit matrix $Z$ as follows:

$$\pi(j|\boldsymbol{w}) := \mathbf{rank}\left( \sum_{t=1}^k z_{jt} \ \middle| \ \left\{ \sum_{t=1}^k z_{1t}, \cdots, \sum_{t=1}^k z_{nt} \right\} \right) = \mathbf{rank}\left( \boldsymbol{h}_j^\top \boldsymbol{w} \mid \{\boldsymbol{h}_1^\top \boldsymbol{w}, \ldots, \boldsymbol{h}_n^\top \boldsymbol{w}\} \right). \tag{3}$$

As mentioned earlier, it is hard to design an ideal candidate screening procedure generating $\mathcal{C}(\boldsymbol{w})$ based on the marginal ranking. Because the main goal for the candidate screening phase is to quickly identify candidates which are highly possible to be top-ranked items, it suffices to have an efficient procedure generating $\mathcal{C}(\boldsymbol{w})$ by an approximation ranking. Here we propose a greedy heuristic ranking:

$$\bar{\pi}(j|\boldsymbol{w}) := \mathbf{rank}\left( \max_{t=1}^k z_{jt} \ \middle| \ \left\{ \max_{t=1}^k z_{1t}, \cdots, \max_{t=1}^k z_{nt} \right\} \right), \tag{4}$$

which is obtained by replacing the summation terms in (3) by $\max$ operators. The intuition behind this heuristic is that the largest element of $\boldsymbol{z}_j$ multiplied by $k$ is an upper bound of $\boldsymbol{h}_j^\top \boldsymbol{w}$:

$$\boldsymbol{h}_j^\top \boldsymbol{w} = \sum_{t=1}^k z_{jt} \leq k \max\{z_{jt} : t = 1, \ldots, k\}. \tag{5}$$

Thus, $\bar{\pi}(j|\boldsymbol{w})$, which is induced by such an upper bound of $\boldsymbol{h}_j^\top \boldsymbol{w}$, could be a reasonable approximation ranking for the marginal ranking $\pi(j|\boldsymbol{w})$.

Next we design an efficient procedure which generates $\mathcal{C}(\boldsymbol{w})$ according to the ranking $\bar{\pi}(j|\boldsymbol{w})$ defined in (4). First, based on the relative orderings of $\{z_{jt}\}$, we consider the *joint ranking* and the *conditional ranking* defined as follows:

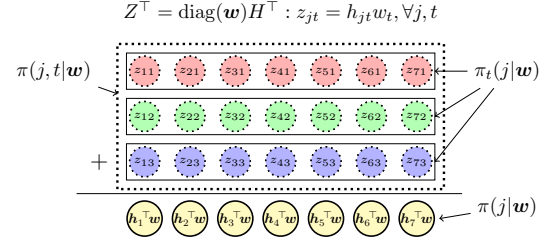

- Joint ranking: $\pi(j,t|\boldsymbol{w})$ is the exact ranking over the $nk$ entries of $Z$.

$$\pi(j,t|\boldsymbol{w}) := \mathbf{rank}(z_{jt} \mid \{z_{11}, \ldots, z_{nk}\}).$$

- Conditional ranking: $\pi_t(j|\boldsymbol{w})$ is the exact ranking over the $n$ entires of the $t$-th row of $Z^\top$.

Figure 1: $nk$ multiplications in a naive linear MIPS approach. $\pi(j,t|\boldsymbol{w})$: joint ranking. $\pi_t(j|\boldsymbol{w})$: conditional ranking. $\pi(j|\boldsymbol{w})$: marginal ranking.

$$\pi_t(j|\boldsymbol{w}) := \mathbf{rank}(z_{jt} \mid \{z_{1t}, \ldots, z_{nt}\}).$$

See Figure 1 for an illustration for both rankings. Similar to the marginal ranking, both joint and conditional rankings are query dependent.

Observe that, in (4), for each $j$, only a single maximum entry of $Z$, $\max_{t=1}^k z_{jt}$, is considered to obtain the ranking $\bar{\pi}(j|\boldsymbol{w})$. To generate $\mathcal{C}(\boldsymbol{w})$ based on $\bar{\pi}(j|\boldsymbol{w})$, we can iterate $(j,t)$ entries of $Z$ in a greedy sequence such that $(j_1, t_1)$ is visited before $(j_2, t_2)$ if $z_{j_1 t_1} > z_{j_2 t_2}$, which is exactly the sequence corresponding to the joint ranking $\pi(j,t|\boldsymbol{w})$. Each time an entry $(j,t)$ is visited, we can include the index $j$ into $\mathcal{C}(\boldsymbol{w})$ if $j \notin \mathcal{C}(\boldsymbol{w})$. In Theorem 1, we show that the sequence to include a newly observed $j$ into $\mathcal{C}(\boldsymbol{w})$ is exactly the sequence induced by the ranking $\bar{\pi}(j|\boldsymbol{w})$ defined in (4).

**Theorem 1.** *For all $j_1$ and $j_2$ such that $\bar{\pi}(j_1|\boldsymbol{w}) < \bar{\pi}(j_2|\boldsymbol{w})$, $j_1$ will be included into $\mathcal{C}(\boldsymbol{w})$ before $j_2$ if we iterate $(j,t)$ pairs following the sequence induced by the joint ranking $\pi(j,t|\boldsymbol{w})$. A proof can be found in Section D.1.*

At first glance, generating $(j,t)$ in the sequence according to the joint ranking $\pi(j,t|\boldsymbol{w})$ might require the access to all the $nk$ entries of $Z$ and cost $O(nk)$ time. In fact, based on Property 1 of conditional rankings, we can design an efficient variant of the $k$-way merge algorithm [8] to generate $(j,t)$ pairs in the desired sequence iteratively.

**Property 1.** *Given a fixed candidate matrix $H$, for any possible $\boldsymbol{w}$ with $w_t \neq 0$, the conditional ranking $\pi_t(j|\boldsymbol{w})$ is either $\pi_{t+}(j)$ or $\pi_{t-}(j)$, where $\pi_{t+}(j) = \mathbf{rank}(h_{jt} \mid \{h_{1t}, \ldots, h_{nt}\})$, and*

$$\pi_{t-}(j) = \mathbf{rank}(-h_{jt} \mid \{-h_{1t}, \ldots, -h_{nt}\}). \text{ In particular, } \pi_t(j|\boldsymbol{w}) = \begin{cases} \pi_{t+}(j) & \text{if } w_t > 0, \\ \pi_{t-}(j) & \text{if } w_t < 0. \end{cases}$$

Property 1 enables us to characterize a *query dependent* conditional ranking $\pi_t(j|\boldsymbol{w})$ by two *query independent* rankings $\pi_{t+}(j)$ and $\pi_{t-}(j)$. Thus, for each $t$, we can construct and store a **sorted** index array $\mathbf{s_t[r]}$, $r = 1, \ldots, n$ such that

$$\pi_{t+}(\mathbf{s_t}[1]) \leq \pi_{t+}(\mathbf{s_t}[2]) \leq \cdots \leq \pi_{t+}(\mathbf{s_t}[n]), \tag{6}$$

$$\pi_{t-}(\mathbf{s_t}[1]) \geq \pi_{t-}(\mathbf{s_t}[2]) \geq \cdots \geq \pi_{t-}(\mathbf{s_t}[n]). \tag{7}$$

Thus, in the phase of *query-independent data structure construction* of Greedy-MIPS, we compute and store $k$ *query-independent* rankings $\pi_{t+}(\cdot)$ by $k$ sorted index arrays of length $n$: $\mathbf{s_t[r]}$, $r = 1, \ldots, n$, $t = 1, \ldots, k$. The entire construction costs $O(kn \log n)$ time and $O(kn)$ space.

Next we describe the details of the proposed Greedy-MIPS algorithm for a given query $\boldsymbol{w}$ and a budget $B$. Greedy-MIPS utilizes the idea of the $k$-way merge algorithm to visit $(j,t)$ entries of $Z$ according to the joint ranking $\pi(j,t|\boldsymbol{w})$. Designed to merge $k$ **sorted** sublists into a single sorted list, the $k$-way merge algorithm uses 1) $k$ pointers, one for each sorted sublist, and 2) a binary tree structure (either a heap or a selection tree) containing the elements pointed by these $k$ pointers to obtain the next element to be appended into the sorted list [8].

### 4.1 Query-dependent Pre-processing

We divide $nk$ entries of $(j,t)$ into $k$ groups. The $t$-th group contains $n$ entries: $\{(j,t) : j = 1, \ldots, n\}$. Here we need an *iterator* playing a similar role as the pointer which can iterate index $j \in \{1, \ldots, n\}$ in the *sorted* sequence induced by the conditional ranking $\pi_t(\cdot|\boldsymbol{w})$. Utilizing Property 1, the $t$-th pre-computed sorted arrays $\mathbf{s_t[r]}$, $r = 1, \ldots, n$ can be used to construct such an iterator, called CondIter, which supports current() to access the currently pointed index $j$ and getNext() to

**Algorithm 1** `CondIter`: an iterator over $j \in \{1, \ldots, n\}$ based on the conditional ranking $\pi_t(j|\boldsymbol{w})$. This code assumes that the $k$ sorted index arrays $\mathtt{s_t[r]}$, $r = 1, \ldots, n$, $t = 1, \ldots, k$ are available.

```
class CondIter:
    def constructor(dim_idx, query_val):
        t, w, ptr ← dim_idx, query_val, 1
    def current():
```
$$\textbf{return} \begin{cases} \mathtt{s_t[ptr]} & \text{if } \mathtt{w} > 0, \\ \mathtt{s_t[n - ptr + 1]} & \text{otherwise.} \end{cases}$$
```
    def hasNext(): return (ptr < n)
    def getNext():
        ptr ← ptr + 1 and return current()
```

**Algorithm 2** Query-dependent pre-processing procedure in Greedy-MIPS.

- **Input:** query $\boldsymbol{w} \in \mathbb{R}^k$
- For $t = 1, \ldots, k$
  - $\mathtt{iters}[t] \leftarrow \mathtt{CondIter}(t, w_t)$
  - $z \leftarrow h_{jt} w_t$, where $j = \mathtt{iters}[t].\mathtt{current}()$
  - $\mathtt{Q.push}((z, t))$
- **Output:**
  - $\mathtt{iters}[t]$, $t \leq k$: iterators for $\pi_t(\cdot|\boldsymbol{w})$.
  - Q: a max-heap of
    $$\left\{ (z, t) \mid z = \max_{j=1}^{n} z_{jt}, \ \forall t \leq k \right\}.$$

advance the iterator. In Algorithm 1, we describe a pseudo code for `CondIter`, which utilizes the facts (6) and (7) such that both the construction and the index access cost $O(1)$ space and $O(1)$ time. For each $t$, we use $\mathtt{iters}[t]$ to denote the `CondIter` for the $t$-th conditional ranking $\pi_t(j|\boldsymbol{w})$.

Regarding the binary tree structure used in Greedy-MIPS, we consider a max-heap Q of $(z, t)$ pairs. $z \in \mathbb{R}$ is the *compared key* used to maintain the heap property of Q, and $t \in \{1, \ldots, k\}$ is an integer to denote the index to a entry group. Each $(z, t) \in$ Q denotes the $(j, t)$ entry of $Z$ where $j = \mathtt{iters}[t].\mathtt{current}()$ and $z = z_{jt} = h_{jt} w_t$. Note that there are most $k$ elements in the max-heap at any time. Thus, we can implement Q by a binary heap such that 1) $\mathtt{Q.top}()$ returns the maximum pair $(z, t)$ in $O(1)$ time; 2) $\mathtt{Q.pop}()$ deletes the maximum pair of Q in $O(\log k)$ time; and 3) $\mathtt{Q.push}((z, t))$ inserts a new pair in $O(\log k)$ time. Note that the entire Greedy-MIPS can also be implemented using a selection tree among the $k$ entries pointed by the $k$ iterators. See Section B in the supplementary material for more details.

In the *query-dependent pre-processing* phase, we need to construct $\mathtt{iters}[t]$, $t = 1, \ldots, k$, one for each conditional ranking $\pi_t(j|\boldsymbol{w})$, and a max-heap Q which is initialized to contain $\left\{ (z, t) \mid z = \max_{j=1}^{n} z_{jt}, \ t \leq k \right\}$. A detailed procedure is described in Algorithm 2 which costs $O(k \log k)$ time and $O(k)$ space.

### 4.2 Candidate Screening

The core idea of Greedy-MIPS is to iteratively traverse $(j, t)$ entries of $Z$ in a *greedy* sequence and collect newly observed indices $j$ into $\mathcal{C}(\boldsymbol{w})$ until $|\mathcal{C}(\boldsymbol{w})| = B$. In particular, if $r = \pi(j, t|\boldsymbol{w})$, then $(j, t)$ entry is visited at the $r$-th iterate. Similar to the $k$-way merge algorithm, we describe a detailed procedure in Algorithm 3, which utilizes the `CondIter` in Algorithm 1 to perform the screening.

Recall both requirements of a viable candidate screening procedure for budgeted MIPS: 1) the flexibility to control the size $|\mathcal{C}(\boldsymbol{w})| \leq B$; and 2) an efficient procedure runs in $O(Bk)$. First, it is clear that Algorithm 3 has the flexibility to control the size of $\mathcal{C}(\boldsymbol{w})$ by the exiting condition of the outer **while**-loop. Next, to analyze the overall time complexity of Algorithm 3, we need to know the number of the $z_{jt}$ entries the algorithm iterates before $\mathcal{C}(\boldsymbol{w}) = B$. Theorem 2 gives an upper bound on this number of iterations.

**Theorem 2.** *There are at least $B$ distinct indices $j$ in the first $Bk$ entries $(j, t)$ in terms of the joint ranking $\pi(j, t|\boldsymbol{w})$ for any $\boldsymbol{w}$; that is,*

$$|\{j \mid \forall (j, t) \text{ such that } \pi(j, t|\boldsymbol{w}) \leq Bk\}| \geq B. \tag{8}$$

A detailed proof can be found in Section D of the supplementary material. Note that there are some $O(\log k)$ time operations within both the outer and inner **while** loops such as $\mathtt{Q.push}((\mathtt{z}, \mathtt{t}))$ and $\mathtt{Q.pop}())$. As the goal of the screening procedure is to identify $j$ indices only, we can skip the $\mathtt{Q.push}((\mathtt{z_{jt}}, \mathtt{t}))$ for an entry $(j, t)$ with the $j$ having been included in $\mathcal{C}(\boldsymbol{w})$. As a results, we can guarantee that $\mathtt{Q.pop}()$ is executed at most $B + k - 1$ times when $|\mathcal{C}(\boldsymbol{w})| = B$. The extra $k - 1$ times occurs in the situation that

$$\mathtt{iters}[1].\mathtt{current}() = \cdots = \mathtt{iters}[k].\mathtt{current}()$$

at the beginning of the entire screening procedure.

To check weather a index $j$ in the current $\mathcal{C}(\boldsymbol{w})$ in $O(1)$ time, we use an auxiliary zero-initialized array of length $n$: `visited`$[j]$, $j = 1, \ldots, n$ to denote whether an index $j$ has been included in $\mathcal{C}(\boldsymbol{w})$ or not. As $\mathcal{C}(\boldsymbol{w})$ contains at most $B$ indices, only $B$ elements of this auxiliary array will be modified during the screening procedure. Furthermore, the auxiliary array can be reset to zero using $O(B)$ time in the end of Algorithm 3, so this auxiliary array can be utilized again for a different query vector $\boldsymbol{w}$. Notice that Algorithm 3 still iterates $Bk$ entries of $Z$ but at most $B + k - 1$ entries will be pushed into or pop from the max-heap `Q`. Thus, the overall time complexity of Algorithm 3 is $O(Bk + (B + k)\log k) = O(Bk)$, which makes Greedy-MIPS a viable budgeted MIPS approach.

---

**Algorithm 3** An improved candidate screening procedure in Greedy-MIPS. The time complexity is $O(Bk)$.

- **Input:**
  - $\mathcal{H}$, $\boldsymbol{w}$, and the computational budget $B$
  - `Q` and `iters`$[t]$: output of Algorithm 2
  - $\mathcal{C}(\boldsymbol{w})$: an empty list
  - `visited`$[j] = 0$, $\forall j \leq n$: a zero-initialized array.
- **While** $|\mathcal{C}(\boldsymbol{w})| < B$:
  - $(z, t) \leftarrow$ `Q.pop()` $\qquad\qquad \cdots O(\log k)$
  - $j \leftarrow$ `iters`$[t]$`.current()`
  - **If** `visited`$[j] = 0$:
    * append $j$ into $\mathcal{C}(\boldsymbol{w})$ and `visited`$[j] \leftarrow 1$
  - **While** `iters`$[t]$`.hasNext()`:
    * $j \leftarrow$ `iters`$[t]$`.getNext()`
    * **if** `visited`$[j] = 0$:
      — $z \leftarrow h_{jt}w_t$ and `Q.push`$((z, t))$ $\cdots O(\log k)$
      — **break**
- `visited`$[j] \leftarrow 0, \forall j \in \mathcal{C}(\boldsymbol{w})$ $\qquad \cdots O(B)$
- **Output:** $\mathcal{C}(\boldsymbol{w}) = \{j \mid \bar{\pi}(j|\boldsymbol{w}) \leq B\}$

---

### 4.3 Connection to Sampling Approaches

Sample-MIPS, as mentioned earlier, is essentially a sampling algorithm with replacement scheme to draw entries of $Z$ such that $(j, t)$ is sampled with the probability proportional to $z_{jt}$. Thus, Sample-MIPS can be thought as a traversal of $(j, t)$ entries using in a stratified random sequence determined by a distribution of the *values* of $\{z_{jt}\}$, while the core idea of Greedy-MIPS is to iterate $(j, t)$ entries of $Z$ in a greedy sequence induced by the *ordering* of $\{z_{jt}\}$.

Next, we discuss the differences of Greedy-MIPS from Sample-MIPS and Diamond-MSIPS. Sample-MIPS can be applied to the situation where both $\mathcal{H}$ and $\boldsymbol{w}$ are nonnegative because of the nature of sampling scheme. In contrast, Greedy-MIPS can work on any MIPS problems as only the ordering of $\{z_{jt}\}$ matters in Greedy-MIPS. Instead of $\boldsymbol{h}_j^\top \boldsymbol{w}$, Diamond-MSIPS is designed for the MSIPS problem which is to identify candidates with largest $(\boldsymbol{h}_j^\top \boldsymbol{w})^2$ or $|\boldsymbol{h}_j^\top \boldsymbol{w}|$ values. In fact, for nonnegative MIPS problems, the diamond sampling is equivalent to Sample-MIPS. Moreover, for MSIPS problems with negative entries, when the number of samples is set to be the budget $B$,[2] the Diamond-MSIPS is equivalent to apply Sample-MIPS to sample $(j, t)$ entries with the probability $p(j, t) \propto |z_{jt}|$. Thus, the applicability of the existing sampling-based approaches remains limited for general MIPS problems.

### 4.4 Theoretical Guarantee

Greedy-MIPS is an algorithm based on a greedy heuristic ranking (4). Similar to the analysis of Quicksort, we study the average complexity of Greedy-MIPS by assuming a distribution of the input dataset. For simplicity, our analysis is performed on a stochastic implicit matrix $Z$ instead of $\boldsymbol{w}$. Each entry in $Z$ is assumed to follow a uniform distribution `uniform`$(a, b)$. We establish Theorem 3 to prove that the number of entries $(j, t)$ iterated by Greedy-MIPS to include the index to the largest candidate is sublinear to $n = |\mathcal{H}|$ with a high probability when $n$ is large enough.

**Theorem 3.** *Assume that all the entries $z_{jt}$ are drawn from a uniform distribution* `uniform`$(a, b)$. *Let $j^*$ be the index to the largest candidate (i.e., $\pi(j^*|Z) = 1$). With high probability, we have* $\bar{\pi}(j^*|Z) \leq \mathcal{O}(k \log(n) n^{\frac{1}{k}})$. *A detailed proof can be found in the supplementary material.*

Notice that theoretical guarantees for approximate MIPS is challenging even for randomized algorithms. For example, the analysis for Diamond-MSIPS in [3] requires nonnegative assumptions and only works on MSIPS (max-squared-inner-product search) problems instead of MIPS problems.

## 5 Experimental Results

In this section, we perform extensive empirical comparisons to compare Greedy-MIPS with other state-of-the-art fast MIPS approaches on both real-world and synthetic datasets: We use netflix and yahoo-music as our real-world recommender system datasets. There are $17,770$ and $624,961$ items in netflix and yahoo-music, respectively. In particular, we obtain the user embeddings $\{\boldsymbol{w}_i\} \in \mathbb{R}^k$

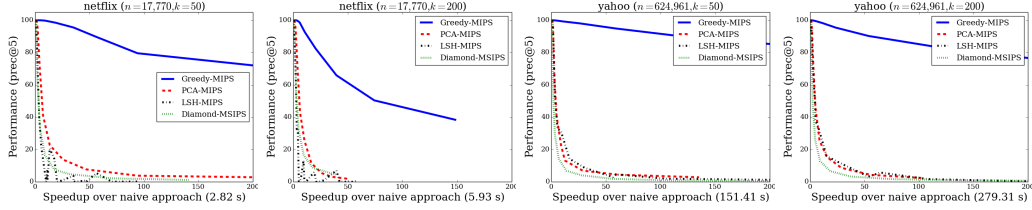

Figure 2: MIPS comparison on netflix and yahoo-music.

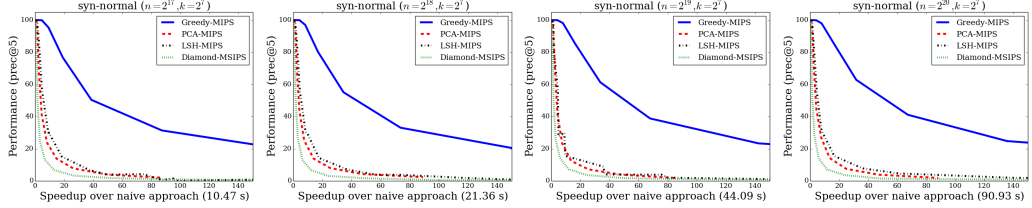

Figure 3: MIPS comparison on synthetic datasets with $n \in 2^{\{17,18,19,20\}}$ and $k = 128$. The datasets used to generate results are created with each entry drawn from a normal distribution.

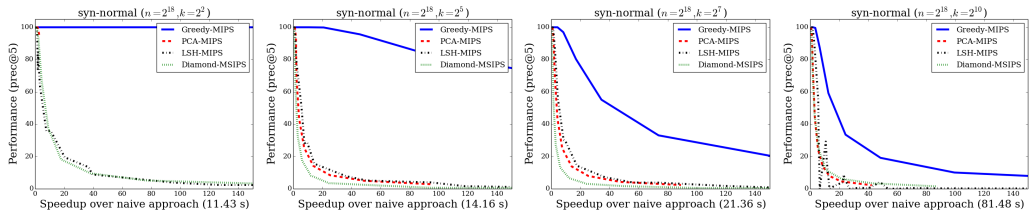

Figure 4: MIPS Comparison on synthetic datasets with $n = 2^{18}$ and $k \in 2^{\{2,5,7,10\}}$. The datasets used to generate results on are created with each entry drawn from a normal distribution.

and item embeddings $\boldsymbol{h}_j \in \mathbb{R}^k$ by the standard low-rank matrix factorization [4] with $k \in \{50, 200\}$. We also generate synthetic datasets with various $n = 2^{\{17,18,19,20\}}$ and $k = 2^{\{2,5,7,10\}}$. For each synthetic dataset, both candidate vector $\boldsymbol{h}_j$ and query $\boldsymbol{w}$ vector are drawn from the normal distribution.

## 5.1 Experimental Settings

To have fair comparisons, all the compared approaches are implemented in C++.

- Greedy-MIPS: our proposed approach in Section 4.
- PCA-MIPS: the approach proposed in [2]. We vary the depth of PCA tree to control the trade-off.
- LSH-MIPS: the approach proposed in [12, 14]. We use the nearest neighbor transform function proposed in [2, 12] and use the random projection scheme as the LSH function as suggested in [12]. We also implement the standard amplification procedure with an OR-construction of $b$ hyper LSH hash functions. Each hyper LSH function is a result of an AND-construction of $a$ random projections. We vary values $(a, b)$ to control the trade-off.
- Diamond-MSIPS: the sampling scheme proposed in [3] for the maximum squared inner product search. As it shows better performance than LSH-MIPS in [3] in terms of MIPS problems, we also include Diamond-MSIPS into our comparison.
- Naive-MIPS: the baseline approach which applies a linear search to identify the exact top-$K$ candidates.

**Evaluation Criteria.** For each dataset, the actual top-20 items for each query are regarded as the ground truth. We report the average performance on a randomly selected 2,000 query vectors. To evaluate the search quality, we use the precision on the top-$P$ prediction (prec@$P$), obtained by selecting top-$P$ items from $\mathcal{C}(\boldsymbol{w})$ returned by the candidate screening procedure. Results with $P = 5$ is shown in the paper, while more results with various $P$ are in the supplementary material. To evaluate the search efficiency, we report the relative speedups over the Naive-MIPS approach:

$$\text{speedup} = \frac{\text{prediction time required by Naive-MIPS}}{\text{prediction time by a compared approach}}.$$

**Remarks on Budgeted MIPS versus Non-Budgeted MIPS.** As mentioned in Section 3, PCA-MIPS and LSH-MIPS cannot handle MIPS with a budget. Both the search computation cost and the search quality are fixed when the corresponding data structure is constructed. As a result, to understand the trade-off between search efficiency and search quality for these two approaches, we can only try various values for its parameters (such as the depth for PCA tree and the amplification parameters $(a, b)$ for LSH). For each combination of parameters, we need to re-run the entire query-independent pre-processing procedure to construct a new data structure.

**Remarks on data structure construction.** Note that the time complexity for the construction for Greedy-MIPS is $O(kn \log n)$, which is on par to $O(kn)$ for Diamond-MSIPS, and faster than $O(knab)$ for LSH-MIPS and $O(k^2 n)$ for PCA-MIPS. As an example, the construction for Greedy-MIPS only takes around 10 seconds on yahoo-music with $n = 624,961$ and $k = 200$.

### 5.2   Experimental Results

**Results on Real-World Data sets.** Comparison results for netflix and yahoo-music are shown in Figure 2. The first, second, and third columns present the results with $k = 50$ and $k = 200$, respectively. It is clearly observed that given a fixed speedup, Greedy-MIPS yields predictions with much higher search quality. In particular, on the yahoo-music data set with $k = 200$, Greedy-MIPS runs 200x faster than Naive-MIPS and yields search results with $p@5 = 70\%$, while none of PCA-MIPS, LSH-MIPS, and Diamond-MSIPS can achieve a $p@5 > 10\%$ while maintaining the similar 200x speedups.

**Results on Synthetic Data Sets.** We also perform comparisons on synthetic datasets. The comparison with various $n \in 2^{\{17,18,19,20\}}$ is shown in Figure 3, while the comparison with various $k \in 2^{\{2,5,7,10\}}$ is shown in Figure 4. We observe that the performance gap between Greedy-MIPS over other approaches remains when $n$ increases, while the gap becomes smaller when $k$ increases. However, Greedy-MIPS still outperforms other approaches significantly.

## 6   Conclusions and Future Work

In this paper, we develop a novel Greedy-MIPS algorithm, which has the flexibility to handle budgeted MIPS, and yields surprisingly superior performance compared to state-of-the-art approaches. The current implementation focuses on MIPS with dense vectors, while in the future we plan to implement our algorithm also for high dimensional sparse vectors. We also establish a theoretical guarantee for Greedy-MIPS based on the assumption that data are generated from a random distribution. How to relax the assumption or how to design a nondeterministic pre-processing step for Greedy-MIPS to satisfy the assumption are interesting future directions of this work.

### Acknowledgements

This research was supported by NSF grants CCF-1320746, IIS-1546452 and CCF-1564000. CJH was supported by NSF grant RI-1719097.

## Footnotes

[2]This setting is used in the experiments in [3].

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
