[Supplementary Material]

# Supplementary Material:
## A Greedy Approach for Budgeted Maximum Inner Products Search

## A  More Details about Existing Approaches for Fast MIPS

### A.1  Approaches with Nearest Neighbor Search Reduction

We briefly introduce the concept of the reduction proposed in [2]. First, we consider the relationship between the Euclidean distance and the inner product:

$$\|\boldsymbol{w} - \boldsymbol{h}_{j_1}\|^2 = \|\boldsymbol{w}\|^2 + \|\boldsymbol{h}_{j_1}\|^2 - 2\boldsymbol{w}^\top \boldsymbol{h}_{j_1}$$
$$\|\boldsymbol{w} - \boldsymbol{h}_{j_2}\|^2 = \|\boldsymbol{w}\|^2 + \|\boldsymbol{h}_{j_2}\|^2 - 2\boldsymbol{w}^\top \boldsymbol{h}_{j_2}.$$

When all the candidate vectors $\boldsymbol{h}_j$ share the same length; that is,

$$\|\boldsymbol{h}_1\| = \|\boldsymbol{h}_2\| = \cdots = \|\boldsymbol{h}_n\|,$$

the MIPS problem is exactly the same as the NNS problem because

$$\|\boldsymbol{w} - \boldsymbol{h}_{j_1}\| > \|\boldsymbol{w} - \boldsymbol{h}_{j_2}\| \iff \boldsymbol{w}^\top \boldsymbol{h}_{j_1} < \boldsymbol{w}^\top \boldsymbol{h}_{j_2} \qquad (9)$$

when $\|\boldsymbol{h}_{j_1}\| = \|\boldsymbol{h}_{j_2}\|$. However, when $\|\boldsymbol{h}_{j_1}\| \neq \|\boldsymbol{h}_{j_2}\|$, (9) no longer holds. See Figure 5(a) for an example where not all the candidate vectors have the same length. We can see that $\boldsymbol{h}_1$ is the candidate vector yielding the maximum inner product with $\boldsymbol{w}$, while $\boldsymbol{h}_2$ is the nearest neighbor candidate.

To handle the situation where candidates have different lengths, [2] proposes the following transform to reduce the original MIPS problem with $\mathcal{H}$ and $\boldsymbol{w}$ in a $k$ dimensional space to a new NNS problem with $\hat{\mathcal{H}} = \left\{ \hat{\boldsymbol{h}}_1, \ldots, \hat{\boldsymbol{h}}_n \right\}$ and $\hat{\boldsymbol{w}}$ in a $k+1$ dimensional space:

$$\hat{\boldsymbol{w}} = [\boldsymbol{w}; 0]^\top,$$
$$\hat{\boldsymbol{h}}_j = \left[ \boldsymbol{h}_j; \sqrt{M - \|\boldsymbol{h}_j\|^2} \right]^\top, \ \forall j = 1, \ldots, n, \qquad (10)$$

where $M$ is the maximum squared length over the entire candidate set $\mathcal{H}$:

$$M = \max_{j=1,\ldots,n} \|\boldsymbol{h}_j\|^2.$$

First, we can see that with the above transform, $\left\| \hat{\boldsymbol{h}}_j \right\|^2 = M$ for all $j$:

$$\left\| \hat{\boldsymbol{h}}_j \right\|^2 = \|\boldsymbol{h}_j\|^2 + M - \|\boldsymbol{h}_j\|^2 = M, \ \forall j.$$

Then, for any $j_1 \neq j_2$, we have

$$\left\| \hat{\boldsymbol{w}} - \hat{\boldsymbol{h}}_{j_1} \right\| < \left\| \hat{\boldsymbol{w}} - \hat{\boldsymbol{h}}_{j_2} \right\|$$
$$\iff M + \|\boldsymbol{w}\|^2 - 2\boldsymbol{w}^\top \boldsymbol{h}_{j_1} < M + \|\boldsymbol{w}\|^2 - 2\boldsymbol{w}^\top \boldsymbol{h}_{j_2}$$
$$\iff \boldsymbol{w}^\top \boldsymbol{h}_{j_1} > \boldsymbol{w}^\top \boldsymbol{h}_{j_2}.$$

With the above relationship, the original $k$-dimensional MIPS problem is equivalent to the transformed $k+1$ dimensional NNS problem. In Figure 5(b), we show the transformed NNS problem for the original MIPS problem presented in Figure 5(a).

In [15], another MIPS-to-NNS reduction has been proposed. The high level idea is to apply a transformation to $\mathcal{H}$ such that all the candidate vectors *roughly* have the same length by appending additional $\bar{k}$ dimensions. In the procedure by [15], all the $\boldsymbol{h}_j$ vectors are assumed (or scaled) to have $\|\boldsymbol{h}_j\| \leq U$, $\forall j$, where $U < 1$ is a positive constant. Then the following transform is applied to reduce the original $k$-dimensional MIPS problem to a new NNS problem with $(k + \bar{k})$-dimensional vectors $\hat{\mathcal{H}}$ and $\hat{\boldsymbol{w}}$ defined as:

$$\hat{\boldsymbol{w}} = [\boldsymbol{w}; \mathbf{0}_{\bar{k}}]^\top$$
$$\hat{\boldsymbol{h}}_j = \left[ \boldsymbol{h}_j; 1/2 - \|\boldsymbol{h}_j\|^{2^1}; 1/2 - \|\boldsymbol{h}_j\|^{2^2}; \ldots; 1/2 - \|\boldsymbol{h}_j\|^{2^{\bar{k}}} \right]^\top, \qquad (11)$$

where $\mathbf{0}_{\bar{k}}$ is a zero vector of dimension $\bar{k}$. Because $U < 1$, [15] shows that with the transform (11), we have $\left\|\hat{\boldsymbol{h}}_j\right\|^2 = \bar{k}/4 + \|\boldsymbol{h}_j\|^{2^{\bar{k}+1}}$, with the second term vanishing as $\bar{k} \to \infty$. Thus, all the candidates $\hat{\boldsymbol{h}}_j$ approximately have the same length. We can see the idea behind (11) is similar to (10): transforming $\mathcal{H}$ to $\hat{\mathcal{H}}$ such that all the candidates have the same length. Note that (10) achieves this goal exactly while (11) achieves this goal approximately. Both transforms show a similar empirical performance in [12].

There are many choices to solve the transformed NNS problem after the MIPS-to-NN reduction has been applied. In [12, 14, 15], various locality sensitive hashing schemes have been considered. In [2], a PCA-tree based approach is proposed, and shows better performance than LSH-based approaches, which is consistent to the empirical observations in [1] and our experimental results shown in Section 5. In [1], a simple K-means clustering algorithm is proposed to handled the transformed NNS problem.

### A.2 Sampling-based Approaches

The idea of the sampling-based MIPS approach is first proposed in [5] as an approach to perform approximate matrix-matrix multiplications. Its applicability on MIPS problems is studied very recently [3]. The idea behind a sampling-based approach called Sample-MIPS, is about to design an efficient sampling procedure such that the $j$-th candidate is selected with probability $p(j)$:

$$p(j) \sim \boldsymbol{h}_j^\top \boldsymbol{w}.$$

In particular, Sample-MIPS is an efficient scheme to sample $(j, t) \in [n] \times [k]$ with the probability $p(j, t)$:

$$p(j, t) \sim h_{jt} w_t.$$

Each time a pair $(j, t)$ is sampled, we increase the count for the $j$-th item by one. By the end of the sampling process, the spectrum of the counts forms an estimation of $n$ inner product values. Due to the nature of the sampling approach, it can only handle the situation where all the candidate vectors and query vectors are *nonnegative*.

Diamond-MSIPS, a diamond sampling scheme proposed in [3], is an extension of Sample-MIPS to handle the maximum **squared** inner product search problem (MSIPS) where the goal is to identify candidate vectors with largest values of $\left(\boldsymbol{h}_j^\top \boldsymbol{w}\right)^2$. If both $\boldsymbol{w}$ and $\mathcal{H}$ are nonnegative or $\boldsymbol{h}_j^\top \boldsymbol{w} \geq 0$, $\forall j$, MSIPS can be used to generate the solutions for MIPS. However, the solutions to MSIPS can be very different from the solutions to MIPS in general. For example, if all the inner product values are negative, the ordering for MSIPS is the exactly reverse ordering induced by MIPS. Here we can see that the applicability of both Sample-MIPS and Diamond-MSIPS to MIPS is very limited.

## B More Details about Greedy-MIPS

### B.1 A Motivating Example for Greedy-MIPS

We demonstrate that an *ideal approach* exists for budgeted MIPS when $k = 1$. It is not hard to observe that Property 2 holds for any given $\mathcal{H} = \{h_1, \ldots, h_n \mid h_j \in \mathbb{R}\}$:

**Property 2.** *For any nonzero query $w \in \mathbb{R}$ and any budget $B > 0$, there are only two possible results for that top $B$ inner products between $w$ and $\mathcal{H}$:*

$$w > 0 \Rightarrow \textit{Largest } B \textit{ elements in } \mathcal{H},$$
$$w < 0 \Rightarrow \textit{Smallest } B \textit{ elements in } \mathcal{H}.$$

This property leads to the following simple approach, which is an ideal procedure for the budgeted MIPS problem when $k = 1$:

- *Query-independent data structure:* a **sorted** list of indices of $\mathcal{H}$: $\mathtt{s[r]}$, $\mathtt{r} = 1, \ldots, n$ such that $\mathtt{s[r]}$ stores the index to the $r$-th largest candidate. That is

$$h_{\mathtt{s[1]}} \geq h_{\mathtt{s[2]}} \geq \cdots \geq h_{\mathtt{s[n]}},$$

- *Candidate screening phase:* for any given $w \neq 0$ and $B > 0$, return

$$\begin{cases} \text{first } B \text{ elements: } \{\mathtt{s[1]}, \ldots, \mathtt{s[B]}\} & \text{if } w > 0, \\ \text{last } B \text{ elements: } \{\mathtt{s[n]}, \ldots, \mathtt{s[n-B+1]}\} & \text{if } w < 0 \end{cases}$$

as the indices of the exact largest-$B$ candidates.

(a) Original MIPS in $\mathbb{R}^2$.　　　　　　　　　(b) Reduced NNS in $\mathbb{R}^3$.

Figure 5: MIPS-to-NN reduction. In 5(a), all the candidate vectors $\{\boldsymbol{h}_j\}$ and the query vector $\boldsymbol{w}$ are in $\mathbb{R}^2$. $\boldsymbol{h}_2$ is the nearest neighbor of $\boldsymbol{w}$, while $\boldsymbol{h}_1$ is the vector yielding the maximum value of the inner product with $\boldsymbol{w}$. In 5(b), the reduction proposed in [2] is applied to $\boldsymbol{w}$ and $\{\boldsymbol{h}_j\}$: $\hat{\boldsymbol{w}} = [\boldsymbol{w}; 0]^\top$ and $\hat{\boldsymbol{h}}_j = [\boldsymbol{h}_j; \sqrt{M - \|\boldsymbol{h}_j\|^2}]^\top$, $\forall j$, where $M = \max_j \|\boldsymbol{h}_j\|^2$. All the transformed vectors are in the 3-dimensional sphere with radius $\sqrt{M}$. As a result, the nearest neighbor of $\hat{\boldsymbol{w}}$ in this transformed 3-dimensional NNS problem, $\hat{\boldsymbol{h}}_1$, corresponds to the vector $\boldsymbol{h}_1$ which yields the maximum inner product value with $\boldsymbol{w}$ in the original 2-dimensional MIPS problem.

Note that for this simple scenario ($k = 1$), neither the *query dependent pre-processing* nor the *candidate ranking* is needed. Thus, the overall time complexity per query is $T_Q = O(B)$. We can see that Property 2 is the key to the correctness of the above procedure. Nevertheless, it is not clear how to generalize Property 2 for MIPS problems with $k \geq 2$. Fortunately, we can directly utilize the fact that Property 2 holds for $k = 1$ to design an efficient greedy procedure for the candidate screening when $k \geq 2$.

## B.2　Greedy-MIPS with a Selection Tree

As there are at most $k$ pairs in the max-heap Q, one from each iters$[t]$, the max-heap can be replaced by a selection tree to achieve a slightly faster implementation as suggested in [8]. In Algorithm 4, we give a pseudo code for the selection tree with a $O(k)$ time constructor, a $O(1)$ time maximum element look-up, and a $O(\log k)$ time updater. To apply the section tree for our Greedy-MIPS, we only need to the following modifications:

- In Algorithm 2, remove Q.push$((z, t))$ from the **for**-loop and construct Q by Q $\leftarrow$ SelectionTree$(\boldsymbol{w}, k, \mathtt{iters})$.
- In Algorithm 3, replace Q.pop() by Q.top() and replace Q.push$((z, t))$ by Q.updateValue$(t, z)$.

(a) At the beginning : $\mathcal{C}(\boldsymbol{w}) = [\,]$

(b) End of iteration-1: $\mathcal{C}(\boldsymbol{w}) = [\mathbf{6}]$

(c) End of iteration-2: $\mathcal{C}(\boldsymbol{w}) = [6, \mathbf{1}]$

(d) End of iteration-3: $\mathcal{C}(\boldsymbol{w}) = [6, 1, \mathbf{7}]$

Figure 6: Illustration of Algorithm 3 with $\boldsymbol{w} = [1, 1, 0.1]^{\top}$ and $B = 3$. The left plot for each sub-figure shows the heap structure in the max-heap Q: the value in each rectangle denotes $z$, and each index $t$ is shown in a different color (red for 1, green for 2, and blue for 3). The sorted index arrays are shown in the upper part of circles on the right plot for each sub-figure; for example, $\mathbf{s}_1[4] = 7$, $\mathbf{s}_2[1] = 6$, and $\mathbf{s}_3[5] = 5$. The value in lower part of circles is the corresponding $h_{jt}$; for example, $h_{71} = -4$, $h_{62} = 7$, and $h_{53} = 29$. Three downward triangles denote the current position of $\texttt{iters}[t]$, $t = 1, 2, 3$. Figure 6(a) shows the status for each data data structure at the beginning of Algorithm 3. Three pairs are pushed into Q: $(-1 = h_{41}w_1, 1)$, $(7 = h_{71}w_2, 2)$, and $(6.9 = h_{13}w_3, 3)$. Figures 6(b)-6(c) show the status in the end of the first and the second iterations of the outer **while**-loop in Algorithm 3. In Figure 6(c), we show that at the third iteration, after $(z, t) = (6, 2) \leftarrow \texttt{Q.pop()}$ is executed and $7 = \texttt{iters}[2].\texttt{current()}$ is appended into $\mathcal{C}(\boldsymbol{w})$, we need to advance $\texttt{iters}[2]$ twice because the index $j = 1$ has been included in $\mathcal{C}(\boldsymbol{w})$. Note that for this example $\boldsymbol{h}_1$ is the candidate with the largest inner product value with $\boldsymbol{w}$.

## C   More Experimental Results

Figure 7: MIPS comparison on netflix and yahoo-music in terms of precision@1.

Figure 8: MIPS comparison on synthetic datasets with $n \in 2^{\{17,18,19,20\}}$ and $k = 128$ in terms of precision@1. The datasets used to generate results are created with each entry drawn from a normal distribution.

**Algorithm 4** A pseudo code of a selection tree used for Greedy-MIPS.

```
class SelectionTree:
    def constructor(w, k, iters):                                    ···O(k)
        K̄ ← min{2^i | 2^i ≥ k}
        for i = 1, ..., 2K̄:
            buf[i] ← (−∞, 0)
        for t = 1, ..., k:
            j ← iters[t].current()
            buf[K̄ + t] ← (h_jt w_t, t)
        for i = K̄, ..., 1:
            if buf[2i].first > buf[2i + 1].first:
                buf[i] ← buf[2i]
            else:
                buf[i] ← buf[2i + 1]
    def top(): return buf[1]                                         ···O(1)
    def updateValue(t, z):                                           ···O(log k)
        i ← K̄ + t
        buf[i] ← (z, t)
        while i > 1:
            i ← ⌊i/2⌋
            if buf[2i].first > buf[2i + 1].first:
                buf[i] ← buf[2i]
            else:
                buf[i] ← buf[2i + 1]
```

Figure 9: MIPS Comparison on synthetic datasets with $n = 2^{18}$ and $k \in 2^{\{2,5,7,10\}}$ in terms of precision@1. The datasets used to generate results on are created with each entry drawn from a normal distribution.

Figure 10: MIPS comparison on netflix and yahoo-music in terms of precision@10.

Figure 11: MIPS comparison on synthetic datasets with $n \in 2^{\{17,18,19,20\}}$ and $k = 128$ in terms of precision@10. The datasets used to generate results are created with each entry drawn from a normal distribution.

Figure 12: MIPS Comparison on synthetic datasets with $n = 2^{18}$ and $k \in 2^{\{2,5,7,10\}}$ in terms of precision@10. The datasets used to generate results on are created with each entry drawn from a normal distribution.

# D Proofs of Theorems

## D.1 Proof of Theorem 1

*Proof.* Let $t_1 = \arg\max_{t=1}^{k} z_{j_1 t}$ and $t_2 = \arg\max_{t=1}^{k} z_{j_2 t}$. By the definition of $t_1$, we have $\pi(j_1, t_1 | \boldsymbol{w}) < \pi(j_1, t | \boldsymbol{w})$, $\forall t \neq t_1$. Thus, $(j_1, t_1)$ will be first entry among $\{(j_1, 1), \dots, (j_1, k)\}$ to be visited in the sequence corresponding to the joint ranking $\pi(j, t | \boldsymbol{w})$. Similarly, $(j_2, t_2)$ will be the first visited entry among $\{(j_2, 1,), \dots, (j_2, k)\}$. We also have

$$\bar{\pi}(j_1 | \boldsymbol{w}) < \bar{\pi}(j_2 | \boldsymbol{w})$$
$$\Rightarrow z_{j_1 t_1} > z_{j_2 t_2}$$
$$\Rightarrow \pi(j_1, t_1 | \boldsymbol{w}) < \pi(j_2, t_2 | \boldsymbol{w}).$$

Thus, $j_1$ will be included into $\mathcal{C}(\boldsymbol{w})$ before $j_2$. $\qquad\square$

## D.2 Proof of Theorem 2

*Proof.* By grouping these first $Bk$ entries by the index $t$ and applying the pigeonhole principle, we know that there exists a group $G$ such that it contains at least $B$ entries. Because each entry in the same group has a distinct $j$ index, we know that the group $G$ contains at least $B$ distinct indices $j$. $\qquad\square$

## D.3 Proof of Theorem 3

**Assumption 1.** *$Z \in \mathbb{R}^{n \times k}$ satisfies $z_{i,j} \sim U[a,b]$. Since we can replace each $z_{i,j}$ by $(z_{i,j} - \frac{a+b}{2})/(b-a)$ without affecting our algorithm, we assume $a = 0, b = 1$ w.l.o.g.*

**Claim 1.** *The probability of picking up the correct candidate with one query is equivalent to:* $p_{n,k} = \Pr(z_{1,1} \geq z_{i,j}, \forall i \in [n], \forall j \in [k] | z_{1,1} \geq z_{1,j}, \forall j \in [k], \& \sum_{j=1}^{k} z_{1,j} \geq \sum_{j=1}^{k} z_{i,j}, \forall i \in [n]) = \Pr(\pi(1|\boldsymbol{z}) = 1 | \pi_1(1|\boldsymbol{z}) = 1, \bar{\pi}(1|\boldsymbol{z}) = 1).$

Since our algorithm Greedy-MIPS only cares about the order of $z_{ij}$, i.e., $\pi_j(i|\boldsymbol{z})$, we could assume w.o.l.g that $\pi_1(1|\boldsymbol{z}) = 1$, which means $z_{11}$ is the largest value in matrix $Z$. Therefore we get the claim.

**Claim 2.** *For any* $a \in [0,1]$,

$$\underbrace{\int_0^a \cdots \int_0^a}_{x_1 + \cdots x_k \geq (k-1)a} (ka - x_1 - \cdots - x_k)^m dx_k \cdots dx_1$$

$$= \frac{a^{m+1}}{m+1} \left( \sum_{i=0}^k \frac{(-a)^i}{\Pi_{j=2}^{k+1}(m+j)} \right) \sim \frac{a^m + 1}{m+1}$$

*Proof of Claim 2.* Denote $s_t = \sum_{i=1}^t x_i, t = 1, 2, \cdots k,$.

$$\underbrace{\int_0^a \cdots \int_0^a}_{s_k \geq (k-1)a} (ka - s_k)^m dx_k \cdots dx_1$$

$$= \int_0^a \int_{a-x_1}^a \cdots \int_{(k-1)a-s_{k-1}}^a (ka - s_k)^m dx_k \cdots dx_1$$

$$= \frac{1}{m+1}\Big(a^{m+1} - $$

$$- \underbrace{\int_0^a \cdots \int_0^a}_{s_{k-1} \geq (k-2)a} ((k-1)a - s_{k-1})^{m+1} dx_{k-1} \cdots dx_1\Big)$$

$$= \cdots$$

$$= \frac{1}{m+1}a^{m+1} - \frac{1}{(m+1)(m+2)}a^{m+2} + \cdots$$

$$-(-a)^{m+k}\frac{1}{(m+1)\cdots(m+k)}$$

$$= \frac{a^{m+1}}{m+1}\left(1 + \sum_{i=1}^k \frac{(-a)^i}{\Pi_{j=2}^{i+1}(m+j)}\right)$$

$$\sim \frac{a^m + 1}{m+1}$$

In the last step, since $m \geq 1, a \leq 1$,

$$\left|\sum_{i=1}^k \frac{(-a)^i}{\Pi_{j=2}^{i+1}(m+j)})\right| < \sum_{i=2}^k \frac{(-1)^i}{i!} < 1/e.$$

Therefore

$$\frac{a^{m+1}}{m+1}\left(1 + \sum_{i=1}^k \frac{(-a)^i}{\Pi_{j=2}^{i+1}(m+j)}\right) \in \frac{a^{m+1}}{m+1}\left[1 - \frac{1}{e}, 1 + \frac{1}{e}\right]$$

$\square$

**Theorem 4.** *With Assumption 1,* Greedy-MIPS *picks up the correct candidate within* $\mathcal{O}(k\log(n)n^{\frac{1}{k}})$ *queries with high probability.*

This theorem is equivalent to Theorem 3.

*Proof of Theorem 4.* Denote $E_i$ as successfully picking up the correct candidate in $i$-th query, and $\bar{E}_i$ is its negate event. Notice $\Pr(E_i | \bar{E}_1, \cdots \bar{E}_{i-1}) \geq \Pr(E_i) \geq \mathcal{O}(\frac{1}{k}n^{-1/k})$. Therefore $\Pr(E_1 \cup E_2 \cup \cdots \cup E_i) \geq 1 - (1 - p_{k,n})^i$ Therefore $\Pr(\bar{E}_1 \cap \bar{E}_2 \cdots \cap \bar{E}_i) \leq (1 - p_{n,k})^i \leq e^{-ip_{k,n}}$.

There exists constant $c$, when $i = c\frac{\log n}{p_{n,k}} = c\log nkn^{-\frac{1}{k}}$,

$$\Pr(\text{Success in } i \text{ queries}) \geq 1 - \mathcal{O}(\frac{1}{n}).$$

$\square$

**Lemma 1.** $p_{n,k} \geq \mathcal{O}\left(\frac{1}{k}\frac{\Gamma(1+\frac{1}{k})\Gamma(n)}{\Gamma(n+\frac{1}{k})}\right) \sim \frac{1}{k}n^{-1/k}$ *as* $n \to \infty$.

*Proof.*

$$
\begin{aligned}
p_{n,k} &= \Pr(\pi_1(1|\boldsymbol{z}) = 1|\pi(1|\boldsymbol{z}) = 1 \& \bar{\pi}(1|\boldsymbol{z}) = 1) \\
&= \frac{\Pr(\pi_1(1|\boldsymbol{z}) = 1 \& \pi(1|\boldsymbol{z}) = 1 \& \bar{\pi}(1|\boldsymbol{z}) = 1)}{\Pr(\pi(1|\boldsymbol{z}) = 1 \& \bar{\pi}(1|\boldsymbol{z}) = 1)}
\end{aligned}
$$

, where $\Pr(\pi(1|\boldsymbol{z}) = 1 \& \bar{\pi}(1|\boldsymbol{z}) = 1) = \frac{1}{kn}$, since $\pi(1|\boldsymbol{z}) = 1)$ and $\bar{\pi}(1|\boldsymbol{z})$ are two independent events with probability $\frac{1}{n}$ and $\frac{1}{k}$ respectively.

Define $l = \sum_{j=1}^{k} z_{1,j}, \bar{l} = l - z_{1,1}$.

$$
\begin{aligned}
&\Pr(A_{n,k}, B_{n,k}) \\
&= \int_0^1 \cdots \int_0^{z_{1,1}} \left[ \underbrace{\int_0^{z_{1,1}} \cdots \int_0^{z_{1,1}}}_{\sum_j z_{2,j} \leq l} 1 dz_{2,k} \cdots dz_{2,1} \right]^{n-1} dz_{1,k} \cdots dz_{1,1} \\
&> \underbrace{\int_0^1 \cdots \int_0^{z_{1,1}}}_{l \geq (k-1)z_{1,1}} \left[ z_{1,1}^k - \underbrace{\int_0^{z_{1,1}} \cdots \int_0^{z_{1,1}}}_{\sum_j z_{2,j} \geq l} 1 dz_{2,k} \cdots d_{2,1} \right]^{n-1} \\
&\quad dz_{1,k} \cdots dz_{1,1} \\
&= \underbrace{\int_0^1 \cdots \int_0^{z_{1,1}}}_{l \geq (k-1)z_{1,1}} \left[ z_{1,1}^k - \frac{(kz_{1,1} - l)^k}{k!} \right]^{n-1} dz_{1,k} \cdots dz_{1,1} \\
&= \int_0^1 \underbrace{\int_0^{z_{1,1}} \cdots \int_0^{z_{1,1}}}_{\bar{l} \geq (k-2)z_{1,1}} \sum_{i=0}^{n-1} z_{1,1}^{k(n-1-i)}(-1)^i \frac{((k-1)z_{1,1}-\bar{l})^{ki}}{(k!)^i} \\
&\quad \times \binom{n-1}{i} dz_{1,k} \cdots dz_{1,1} \\
&= \int_0^1 \sum_{i=0}^{n-1} z_{1,1}^{k(n-1-i)} \frac{(-1)^i}{(k!)^i} \binom{n-1}{i} \left( \sum_{j=0}^{k-1} \frac{z_{1,1}^j}{\Pi_{p=2}^{j+1}(ki+p)} \right) dz_{1,1} \\
&\sim \sum_{i=0}^{n-1} \frac{(-1)^i}{(ki+1)(kn-k+2)} \binom{n-1}{i}(k!)^{-i}
\end{aligned}
$$

The last two equations are from Claim 2.

Therefore $p_{n,k} \sim \sum_{i=0}^{n-1} \frac{(-1)^i}{ki+1} \binom{n-1}{i}(k!)^{-i}$

Let $f_n(x) = \sum_{i=0}^{n} (-1)^i x^{ki} \binom{n}{i} \frac{1}{ki+1}$. Therefore $p_{n,k} = f_{n-1}((k!)^{-1/k})$.
Then since
$$
\frac{\partial}{\partial x}\left( x f_n(x) \right) = \sum_{i=0}^{n} (-1)^i x^{ki} \binom{n}{i} = (1 - x^k)^n
$$

Therefore $p_{n,k} \sim \int_0^{(k!)^{-1/k}}(1-x^k)^{n-1}dx$ Notice $(k!)^{-1/k} \leq (k^k)^{-1/k} = 1/k$, therefore $\int_0^{(k!)^{-1/k}}(1-x^k)^{n-1}dx \geq \int_0^{1/k}(1-x^k)^{n-1}dx \geq 1/k\int_0^1(1-x^k)^{n-1}dx \sim \frac{\Gamma(1+\frac{1}{k})\Gamma(n)}{\Gamma(n+\frac{1}{k})}$

$\square$