[Reviews · NeurIPS 2017]

Reviewer 1



The aim of the paper is to propose a new greedy approach for Maximum Inner Product Search problem: given a candidate vector, retrieve a set of vectors with maximum inner product to the query vector. This is a crucial step in several machine learning and data mining algorithms, and the state of the art methods work in sub-linear time recently. The originality of the paper is to study the MIPS problem under a computational budget. The proposed approach achieves better balance between search efficiency and quality of the retrieved vectors, and does not require a nearest neighbor search phase, as commonly done by state of the art approaches. The authors claim impressive runtime results (their algorithm is 200x faster than the naive approach), and a top-5 precision greater than 75%. The paper is very dense (the space between two lines seems smaller than the one in the template). However, the paper is well-written and the procedure is well-explained. The proposed method seems also quite original, and comes with theoretical guarantees. The technical results seem sound. Some remarks: Figure 1 should be placed at the top of P.5, it is a bit difficult to follow without the later explanations. The bound used in P.4 needs to be more studied, in order to find, for instance, some properties (or better, an approximation). This bound is a key point of this procedure, and it is used at the beginning. P.5 "visit (j,t) entries of Z": (j,t) is a cell in the matrix Z, however you consider this notation as a number. Maybe "j \times t" entries? The reviewer would be interested to have access to the source code of the algorithm and the data, so as he can reproduce the expriments?

Reviewer 2



The papers proposes a greedy approach based to some sorting among the columns of the matrix of candidates to solve the MIPS problem. Some analysis of the proposed solution is made, this leads to an efficient implementation and an asymptotic bound on the error is provided on a simplified case. The analysis does seems very sharp as the provided bound is sublinear but yields better results for high dimensional vectors (this is likely to be an artifact of the uniform iid assumption over the entries). The proposed idea is nice and simple but the writing makes the paper harder to follow than it could be. The MIPS problem received a lot of attention from the community and the experimental part does not compare to the most recent approaches as ALHS (NIPS'14) nor "Learning and Inference via Maximum Inner Product Search" (ICML'16) which can be considered as an issue. The authors discards theses approaches because they consider that the computational budget cannot as easily controlled than with their approach, but in my opinion this reason is not strong enough to not report their performance on the figures. Moreover the time required to build the indexes is not reported in the time comparisons which can be acceptable when the number of user is very large in front of the number of items but again not very fair in terms of time comparisons. From a practitioner point of view, MIPS is not widely used in production because it remains too slow for large scale systems (even when using dense embeddings). One tend to prefer the use of hashtables to some set of products with some well chosen keys.

Reviewer 3



The paper considers the problem of Maximum Inner Product Search, which is an important retrieval problem for recommender systems task (among others), e.g. find an item which is most similar to a user's preference vector. I'm not particularly familiar with related work on this topic (such as Budgeted MIPS), so had to learn as I went. Essentially (as I understand it) Budgeted MIPS has a trade-off between latency and search quality. The idea here is to greedily develop algorithms that dynamically adjust this tradeoff to improve performance. The actual procedure, as I understand it, seems fairly straightforward. And plenty of detail is given that what's proposed could easily be re-implemented. Even so, the method (as far as I can tell) is novel and appears to be backed up by some theoretical analysis that demonstrates its validity. The analysis itself is a bit limited, in the sense that it makes strong assumptions about the data distribution (uniformity of the vectors). I can imagine several applications where this would not be realistic. However it seems realistic enough for applications like retrieval in a recommender system, which is an important enough application in and of itself. The experiments on real-world datasets seems to confirm that the proposed approach does indeed result in substantial speed increases.